Germination characteristics among different sheepgrass (Leymus chinensis) germplasm during the seed development and after-ripening stages

Yang Weiguang 1 2 3
Liu Shu 1 2
Yuan Guangxiao 1 2
Liu Panpan 1 2
Qi Dongmei 1
Dong Xiaobing 1
Liu Hui 1
Liu Gongshe 1 liugs@ibcas.ac.cn
Li Xiaoxia 1 lixx2013@ibcas.ac.cn
1 Key Laboratory of Plant Resources, Institute of Botany, The Chinese Academy of Sciences , Beijing , China
2 University of Chinese Academy of Sciences , Beijing , China
3 Institute of Animal Sciences in Heilongjiang province , Qiqihar , China
Van de Peer Yves
Electronic publication date: 2019 Apr 10
Publication date: 2019
Volume: 7
Electronic Location ID: e6688
Received 2018 Oct 25; Accepted 2019 Feb 26
Copyright: © 2019 Yang et al.
Copyright year: 2019
Copyright holder: Yang et al.
License: This is an open access article distributed under the terms of the Creative Commons Attribution License, which permits unrestricted use, distribution, reproduction and adaptation in any medium and for any purpose provided that it is properly attributed. For attribution, the original author(s), title, publication source (PeerJ) and either DOI or URL of the article must be cited.
License URL: https://creativecommons.org/licenses/by/4.0/

Keywords: Sheepgrass, Seed development, After-ripening, Seed germination, Germplasm

Funding: Science and Technology Major Project of Inner Mongolia Autonomous Region of China, and the National Key R&D Program of China 2018 YFD1001003, 2016YFC0500704 This study was supported by the Science and Technology Major Project of Inner Mongolia Autonomous Region of China, and the National Key R&D Program of China (Grant 2018 YFD1001003, 2016YFC0500704). The funders had no role in study design, data collection and analysis, decision to publish, or preparation of the manuscript.

==============================
Sheepgrass (Leymus chinensis (Trin.) Tzvel) is an important forage grass in the Eurasian steppe. However, little information is available concerning its seed morphological features and germination characteristics during seed development and after-ripening among different germplasm. To clarify the appropriate seed harvest time and the effects of germplasm, seed development and after-ripening on seed germination, 20 germplasm of sheepgrass were selected. Moreover, the seed morphological and physical changes as well as the seed germination and dormancy characteristics of sheepgrass during seed development stages were analyzed using a seven—d gradient of day after pollination (DAP). The results indicated that the seed water content decreased significantly during 35–42 DAP and that the highest seed germination rate of most germplasm was observed at 35–42 DAP. Thus, 35–42 DAP may be the best time to harvest sheepgrass to obtain the maximum seed germination rate and avoid seed shattering. Furthermore, our results indicated that there were six types of germination patterns, including germplasm with increasing germination rates in the developing seed, such as S19 and S13, and germplasm that maintained a consistently low germination rate, such as S10. Moreover, we compared the seed germination rate of eight germplasm during seed development in both 2016 and 2017, and the results indicated that the seed germination patterns of the eight germplasm were highly consistent between the two consecutive years, suggesting that germplasm rather than year is the major factor in determining germination during seed development. The effect of after-ripening on seed germination was different among the germplasm where four types of germination patterns were revealed for 10 germplasm and resulted in various dormancy features. A two-factor ANOVA analysis suggested that the germplasm of the sheepgrass has a large influence on seed germination, whether during seed development or after-ripening. Thus, these findings lay the foundation for future studies on seed dormancy and germination and may guide the breeding of new cultivars of sheepgrass with better germination performance.

Introduction

Sheepgrass (Leymus chinensis (Trin.) Tzvel) is a perennial species of Poaceae that is widely distributed in the eastern part of the Eurasian steppe, the northern and eastern parts of the People’s Republic of Mongolia, and the northern and northeastern grasslands of China (Lin et al., 2011). As the dominant species in the region and one of the most promising forage grasses, sheepgrass is well known for its abundant foliage, high palatability and high tolerance to various environmental stresses, including arid, saline, and alkaline soils (Lin et al., 2011; Liu & Qi, 2004; Xu & Zhou, 2005). Therefore, sheepgrass is considered to be valuable in both animal husbandry and grassland restoration, and there has been a great demand for high quality sheepgrass seeds in recent decades (Liu & Li, 2015). However, seed germination of sheepgrass is relatively low and unstable due to strong seed dormancy, which limits seed utilization and initial grassland establishment.

Previous studies on the seed dormancy and germination of sheepgrass have mostly focused on dormancy-breaking methods and the environmental and hormonal effects on germination (He et al., 2016; Zhang et al., 2006). The mechanical restriction of the lemma is considered to be the major cause of sheepgrass seed dormancy because removing the lemma can significantly increase germination (Ma et al., 2008). Furthermore, it was reported that the seed dormancy of sheepgrass was significantly affected by the lemma, palea, pericarp/testa, and endosperm of the seed, and their average contribution to seed dormancy was 23.4%, 6.2%, 28.4%, and 42.0%, respectively (He et al., 2016). However, the ecological and physiological role of the lemma was revealed; it plays important roles in water uptake, dehydration and salt tolerance during germination (Lin et al., 2016a). In studies, the dormancy-breaking methods for sheepgrass seed have varied from low temperature treatments to chemical agents, including PEG and GA3 (Gu, Yi & Holubowiecz, 2005; Liu, Wang & Li, 2002; Zhang et al., 2006). Multiple methods to break seed dormancy have been developed; for instance, a combination of seed soaking treatments, which was to presoak the seed in distilled water for 1 day and then soak it in 30% NaOH for 1 h, followed by treatment with 300 μM GA3, was recently reported as an optimum dormancy-breaking method (He et al., 2016). Temperature and plant hormones are considered crucial factors in the regulation of the germination of sheepgrass. Plant hormones, including gibberellin (GA) and abscisic acid (ABA), are involved in seed dormancy and seed germination in many studies (Brady & McCourt, 2003; Kermode, 2005). Previous studies have shown that conditions with fluctuating temperatures are optimal for germination and dormancy-breaking in sheepgrass, as several studies have reported that the germination rate of sheepgrass could be maximized with temperature sets of 20/30 °C (Hu, Huang & Wang, 2012) or 16/28 °C (Yang, Liu & Yuan, 2018). It has also been indicated that a change in temperature affects germination by contributing to GA biosynthesis during seed imbibition (Hu, Huang & Wang, 2012). However, previous studies on the dormancy and germination of sheepgrass seeds mainly examined the mature seeds of very few germplasm and the germination characteristics during seed development and after-ripening are still unclear.

Seed development begins with a double fertilization process, after which the egg cell develops into the embryo and the polar nuclei develop into the endosperm (Goldberg, De Paiva & Yadegari, 1994). In monocots such as rye grass and cereal crops that are covered by the lemma, the caryopsis consists of the embryo, endosperm, and pericarp, where endosperm occupies the most space in the mature seed (James, Denyer & Myers, 2003). During the seed development of rice, it was reported that embryogenesis is completed at approximately six DAF (days after fertilization) and that the seed filling process occurs at six DAF until 20 DAF (Xu et al., 2008). In wheat, which spends relatively more time in the seed developing stage, the seed filling process begins at approximately eight DAF and lasts until 28 DAF (Leprince et al., 2017; Nasehzadeh & Ellis, 2017). Since each part of the developing seed structure contributes to seed dormancy in sheepgrass, embryogenesis and endosperm development and the afterward late maturation could critically determine the seed germination characteristics (Bewley & Black, 1994; Hu, Huang & Wang, 2012).

After the last phase of seed development and maturation, harvesting seed at the proper stage of maturity is of practical importance in seed production (Finch-Savage & Bassel, 2016). Early harvesting may lead to low seed vigor and poor establishment because of an insufficient level of maturation, while delaying harvest can cause seed loss under unfavorable environmental conditions (Leprince et al., 2017). It was recently reported that the optimum harvest time for sheepgrass seed is approximately 39 days after peak anthesis at which time the maximum seed quality, including seed weight, and the ability to germinate have been acquired (Lin et al., 2016a). However, the flowering time of sheepgrass lacks uniformity, varying among cultivars and even batches, which may considerably affect practical seed harvesting. In addition, exposing harvested seed to warm and dry conditions may result in a progressive loss of dormancy, which is commonly known as seed after-ripening, and the change in seed vigor after harvest remains largely undetermined (Baskin & Baskin, 2014; Foley, 2008). Studies on the effect of after-ripening on promoting seed germination had have been conducted in several species, including rice (Veasey et al., 2004), Stipa (Zhang et al., 2017), and Physaria (Cruz, Romano & Dierig, 2012), indicating that the speed and extent of after-ripening can vary among species as well as different cultivars (Baskin & Baskin, 2014; Veasey et al., 2004). After-ripening of the mature seed is practically significant for plantation and seed production, but little has been revealed on the after-ripening effect on harvested seeds of sheepgrass.

Variations in seed dormancy exist in crop germplasm as an adaptation strategy to regulate germination timing (Gu, Chen & Foley, 2003). The traits related to dormancy are considered to be associated with multiple quantitative trait loci (QTL) in barley, wheat, and rice (Gu et al., 2015). Furthermore, study in hybrid wildryes (Leymus) indicated the pleiotropic QTL on Leymus LG6a with major effects on both seed shattering and seed dormancy (Larson & Kellogg, 2009). However, the variations in seed dormancy and germination among cultivars of sheepgrass remain largely unknown (Zhao et al., 2019). In recent years, several new varieties of sheepgrass, for instance the “Zhongke” series, have been identified as good candidates for sheepgrass production (Liu & Li, 2015). Considering the potential value of multiple varieties, newly cultivated and conserved germplasm require further evaluation, which may lead to a comprehensive understanding of the seed dormancy and germination characteristics of sheepgrass. Therefore, based on substantial cultivars of conserved or previously bred sheepgrass, we aimed to (i) investigate the morphological and physiological characteristics of sheepgrass during seed development and (ii) evaluate the variations in seed dormancy and germination among different germplasm of sheepgrass during seed development as well as the after-ripening process in this study.

Materials and Methods

Seed collection

All the involved germplasm of sheepgrass were conserved in the Laboratory of Genetic Resources of the Sheepgrass Research Group at Institute of Botany, Chinese Academy of Sciences, Beijing, China. A total of 20 germplasm of sheepgrass, coded as K8, C5, C1, Z37, C54, S17, S10, S2, C19, S16, Q6, N4, N14, C18, S3, N2, N15, K4, S13, and S19, were selected for the investigation on seed germination during seed development and seed after-ripening. All seeds were harvested under strict hybridization conditions within each germplasm in both 2016 and 2017.

Morphological and physical observations of the developing seed

During seed development after flowering, a total of more than 100 spikelets from different plants of K4, S19, S10, and N14 were collected, dissected and photographed using a digital camera (Canon EOS 6, Canon Inc, Japan) at different days after pollination (DAP). In addition, the average fresh weight and dry weight of the seeds of K4, S19, S10, and N14 at different DAP were tested at 7 DAP, 14 DAP, 21 DAP, 28 DAP, 35 DAP, and 42 DAP from May to June 2016. The seed water content was calculated based on the fresh and dry weight of the seeds. Three replicates of 50 seeds were performed for each germplasm.

Germination test

The germination type of sheepgrass is hypogeal. The germination rate during seed development was tested in 20 germplasm of sheepgrass at 7 DAP, 14 DAP, 21 DAP, 28 DAP, 35 DAP, and 42 DAP. Harvested at 42 DAP, the seeds of all germplasm were stored in darkness at room temperature with a relative humidity of 30% as the after-ripening treatment. To determine the after-ripening effect, the germination rate of ten germplasm was tested at 0 days after harvest (DAH), 15 DAH, 30 DAH, and 60 DAH, while the zero DAH treatments were harvested at 42 DAP. Three replicates of 30 seeds were used for each germplasm at the stage of seed development and during the after-ripening process. The seeds were selected and placed in nine-cm Petri dishes on two layers of moistened filter paper. The dishes were placed in an incubator for 18 days under an alternating temperature controlled system of 28 °C for 12 h and 16 °C for 12 h. The relative humidity in the incubator was maintained at 35% and light at 500 μmol·m−2s−1 was used during the high temperature period each day. Distilled water was added to maintain the moisture content of the Petri dishes as necessary. A seed was considered to have germinated when the emerged radical length reached at least two mm (http://www.seedtest.org/). The percentage germination was calculated as the percentage of the seeds that germinated within the incubation period.

Data analysis

The experimental designs were of a simple random design, and the data were statistically analyzed using SPSS 17.0. Two-way analysis of variance (ANOVA) was performed to test the effects of germplasm and developmental stage, germplasm and year, germplasm and after-ripening time on seed germination. Duncan’s test was used for multiple comparisons among the different treatments.

Results

Morphological changes during seed development in sheepgrass

To obtain the basic characteristics of sheepgrass seed development, we closely observed the morphological features of the developing seeds from flowering until maturity. The structure of a single flower of sheepgrass includes the lemma, the palea, the three stamens and the pistil (Fig. 1A). During the flowering process of sheepgrass, the two flowers near the rachilla blossomed first, and then the other flowers opened on the spikelet. After the anthers were confided in the lemma during five to seven DAP, they started to fall. The fertilized ovary began to expand at three to four DAP, and the spikelet fully expanded. Starch was deposited in the grain, and the inside of the grain was milk white around eight DAP. As the starch accumulated, the inner content of the seed became a white emulsion and became thicker at 16 DAP. Later, as the seed continued to develop and mature, the seed coat began to gain color until the wax inside the seed turned into lumps. At approximately 20 DAP, the seed began to accumulate pigment near the embryo end, and the seed coat gradually turned dark purple or yellow, subsequently turning harder. Furthermore, the palea turned yellow at both ends and became tightly bound to the caryopsis, at which point it was difficult to entirely remove the palea from the caryopsis. Next, the spikelet began to become dehydrated and gradually turned yellow, at which time the seed was considered physiologically mature. The color of the seed coat was mainly observed as dark purple or yellow at approximately 26 DAP, while the palea basically started turning yellow or brown (Fig. 1B), and a population photo of sheepgrass at seed maturity period was in Fig. S1. In addition, it was observed that the flowering time of sheepgrass varied among the germplasm, and based on the flowering time, the 20 germplasm of sheepgrass were classified into three groups: the early, the middle and the late flowering types (Table 1).

Figure 1 Morphology and development of sheepgrass seeds.

(A) The blooming flower of sheepgrass: an intact flower includes the lemma, the palea, the three stamens, and the pistil. (B) The morphology of the developing sheepgrass seed after flowering until seed maturation: The fertilized ovary began to expand at approximately four DAP. As the reserves started to accumulate at eight DAP, the seed reached the milk-ripe stage. At 16 DAP, the seed coat began to gain color, and at approximately 20 DAP, the seed coat began to gradually turn yellow or dark purple and subsequently hardened. The seed coat became fully colored at 26 DAP. Photograph by Gongshe Liu, Xiaoxia Li.

Table 1 Flowering time of experimental cultivars of sheepgrass.

	Flowering time	Germplasm	
Early type	May 5–May 10	K8, C5, C1, Z37, C54, S17	
Mid type	May 11–May 15	S10, S2, C19, S16, Q6, N4, N14	
Late type	May 16–May 20	C18, S3, N2, N15, K4, S13, S19	
Note:

We classified the 20 cultivars of sheepgrass into three types according to the time of flowering. The six cultivars were considered as the early type which flowered on May 5–May 10 in 2016; the flowering time of the mid type was May 11–May 15, involving the seven germplasm; other seven cultivars which flowered on May 16–May 20 were classified as the late type.

Physical changes during seed development in sheepgrass

To investigate the physiological changes during seed development of sheepgrass, we measured the reserve accumulation and water loss at the seed developmental stages of 7 DAP, 14 DAP, 21 DAP, 28 DAP, 35 DAP, and 42 DAP. The results showed that the seed dry weight rapidly increased until 7–14 DAP. After 14 DAP, the increase in seed dry weight slowed and then the dry weight generally reached a maximum at 28 DAP. During seed development and maturation, the water content of the seed continued to decrease after seven DAP and began to rapidly decrease after 28 DAP. The water content of the seed was reduced to under 15% at approximately 35 DAP. Afterward, water loss from the seed continued in late maturation, and the water content fell to approximately 10% at 42 DAP (Fig. 2).

Figure 2 Change in the weight and water content of the developing seeds of sheepgrass.

The results showed that the accumulation of reserves occurred mainly before 14 DAP and that the seed water content decreased rapidly after 28 DAP. Bars represent the mean ± S.E. (n = 4).

Effects of the developmental stages on the seed germination of different germplasm

To further investigate the effect of germplasm on the seed germination of sheepgrass, we randomly selected 20 germplasm according to their flowering time (Table 1). During seed development, the germination rate of each germplasm was tested at 7 DAP, 14 DAP, 21 DAP, 28 DAP, 35 DAP, and 42 DAP. The results showed that the seeds of most of the germplasm were unable to germinate at seven DAP and that only three germplasm, C1, C5, and S17, could germinate at this time with germination rates of 1.67%, 5.83%, and 0.83%, respectively. At 14 DAP, all the germplasm excluding N15 appeared to be capable of germinating (Fig. 3). The results of the germination test after 14 DAP until 42 DAP showed multiple types of variation in germination during seed development in different sheepgrass germplasm. We classified the germination patterns of the developing seeds into the following six types: (1) the germination rate continued to rise gradually with seed development, which included S13, S17, S19, N2, N15, and N14 (Fig. 4A); (2) the germination rate continued to increase at the early developmental stages until 35 DAP and then began to decrease, such as C54, Z37, and C18 (Fig. 4B); (3) the germination rate first decreased from 14 DAP to 21 DAP and then began to increase after 21 DAP, but it then appeared to fall again after 35 DAP, for example, C19 and K4 (Fig. 4C); (4) a zigzag pattern of changes in germination was observed for S2 and C6, in which a varying trend was observed at each of the four developmental stages (Fig. 4D); (5) the germination rate first increased after 14 DAP until 21 DAP, then continued to fall until 35 DAP, and then began to increase again after 35 DAP, which only occurred for S3 (Fig. 4E); and (6) the germination rate was relatively low at all of the developmental stages, including C1, C5, K8, and S10 (Fig. 4F). In addition, a two-way ANOVA was performed, and the results showed that the germination rate of the developing seed was significantly affected by both the germplasm and the developmental period (P < 0.01) (Table 2).

Figure 3 Six types of germination patterns during seed development of different sheepgrass germplasm.

(A) Continuously increasing the germination rate during development, which included the germplasm S13, S17, S19, N2, N15, and N14. (B) Continuously increasing germination until 35 DAP and afterward decreasing, this included the germplasm C54, Z37, and C18. (C) Increasing from 21 DAP to 35 DAP, which included the germplasm C19 and K4. (D) Increasing from 14 DAP to 21 DAP and from 28 DAP to 35 DAP, which included the germplasm S2 and Q6. (E) Increasing from 14 DAP to 21 DAP and from 35 Dap to 42 DAP, which included the germplasm S3. (F) Maintained a low germination rate during development, which included the germplasm C1, C5, K8, and S10. Bars represent the mean ± S.E. (n = 4).

Figure 4 The seed germination patterns of eight germplasm at different developmental stages during two consecutive years.

(A) Seed germination at 14 DAP. (B) Seed germination at 28 DAP. (C) Seed germination at 42 DAP. Different letters indicate significant differences between the 2 years. Bars represent the mean ± S.E. (n = 4).

Table 2 Two-factor analysis of variance of germplasm and developing stage.

Variance source	SS	df	MS	F	P-value	
Germplasm	21,573.19	19	1,135.431	4.371028	1.94E-06	
Developing stage	24,106.84	4	6,026.71	23.2008	1.48E-12	
Error	19,741.99	76	259.763			
Sum	65,422.02	99				
Notes:

df, degree of freedom; MS, mean square; F, F-ratio.

P < 0.05 means significant difference, P < 0.01 means extremely remarkable difference, E is a mathematic constant.

Effects of two consecutive years on the seed germination of different germplasm

Considering the multiple effects on germination characteristics, the germination rates of the developing seeds of contrasting varieties of sheepgrass germplasm were consecutively recorded in 2 years. To determine the genetic regularities of the germination characteristics of different sheepgrass germplasm, we selected eight germplasm and analyzed seed germination during seed development in both 2016 and 2017. The results showed that the seed germination patterns of each germplasm at the different developing stages (14, 28, and 42 DAP) were highly consistent between the two consecutive years (Fig. 4). To further investigate the influence of germplasm and year on the seed germination of sheepgrass, a two-factor ANOVA was performed, which indicated that the germplasm had a significant influence on the germination characteristics of the developing seeds (P < 0.01) and that the seed germination rate of the developing seeds was not significantly different between the 2 years (P > 0.05) (Table 3). The results showed that the germination characteristics of the developing sheepgrass seeds remained relatively constant during the two different years, suggesting that germplasm may significantly affect the seed germination rate of sheepgrass.

Table 3 Two-factor analysis of variance on the seed germination in different developing stages, 14 DAP, 28 DAP, and 42 DAP, of the two consecutive years and different sheepgrass germplasm.

	Variance source	SS	df	MS	F	P	F crit	
14 DAP	Germplasm	2,047.276	7	292.468	75.70445	4.59e−6	3.787044	
Year	5.183211	1	5.183211	1.341658	0.284728	5.591448	
28 DAP	Germplasm	7,756.589	7	1,108.084	136.0715	6.09e−7	3.787044	
Year	8.1225	1	8.1225	0.997434	0.351197	5.591448	
42 DAP	Germplasm	11,816.42	7	1,688.06	28.85806	0.00012	3.787044	
Year	34.63976	1	34.63976	0.592181	0.466746	5.591448	
Notes:

df, degree of freedom; MS, mean square; F, F-ratio.

P < 0.05 means significant difference, P < 0.01 means extremely remarkable difference, e is a mathematic constant.

Effects of after-ripening on the seed germination of different germplasm

To determine the after-ripening effect on the seed germination of sheepgrass, the seed germination rate of 10 sheepgrass germplasm were determined at 0 days, 15 days, 30 days, and 60 days after harvest. Our results showed that the germination rate of the germplasm C5, S3, and N14 tended to significantly increase during after-ripening and that the germination rate reached a peak 30 days after harvest but then dropped significantly 60 days after harvest (Fig. 5A). In contrast, the germplasm S17, C18, and S19 showed a rather high germination rate when first harvested, but after 15 days of after-ripening, the germination rate was significantly reduced; however, the germination rate rose again after 30 days of after-ripening (Fig. 5B). Compared to the other various patterns in the germination rate during the seed after-ripening process, germplasm Z37 showed a continuous significant increasing germination rate after harvest. Similarly, no significant decrease was observed in the seed germination rate of S10 and C19 after harvest (Fig. 5C). However, the germination rate of K4 continued to decrease during the after-ripening process (Fig. 5D). Furthermore, we analyzed the effects of germplasm and after-ripening time on seed germination during after-ripening, and the result of the two-factor ANOVA indicated that the variation by germplasm was greater than that by the after-ripening time (Table 4).

Figure 5 After-ripening effect on the seed germination rates of different sheepgrass germplasm after harvest.

(A) Germination rate of the germplasm C5, N14, S3 continued to increase until 30 days after harvest and then tended to decrease. (B) The germination rate of germplasm S17, C18, and S19 began to increase at 15 days after harvest. (C) The germination rates of the germplasm Z37 and C19 continued to increase after harvest, and the germination rate of S10 did not decrease during the after-ripening process. (D) No increase in germination rate was observed in germplasm K4 after harvest. Bars represent the mean ± S.E. (n = 4).

Table 4 Two-factor analysis of variance of germplasm and after-ripening time.

Variance source	SS	df	MS	F	P-value	F crit	
Germplasm	22,412.84	9	2,490.316	18.22975	2.72E-09	2.250131	
After-ripening time	1,909.136	3	636.3786	4.658455	0.009478	2.960351	
Error	3,688.395	27	136.6072				
Sum	28,010.37	39					
Notes:

df, degree of freedom; MS, mean square; F, F-ratio.

P < 0.05 means significant difference, P < 0.01 means extremely remarkable difference, E is a mathematic constant.

Discussion

For a long time, the cultivation goals for wild species have been to increase seed yield, improve fertility, make the flowering time adapt to the local climate, and reduce seed shattering and dormancy (Konishi et al., 2006). During seed development and maturation, the morphological analyses of seeds have important practical values in the evaluation of germplasm, the cultivation of new varieties, and seed quality monitoring throughout seed production (Bewley & Black, 1994). However, the seed development process of sheepgrass requires more specific morphological and physiological studies due to the limited available information. Seed shattering before harvest significantly reduces seed yield, and sheepgrass is prone to seed shattering; however, the seed does not fall off immediately when it matures (Liu, Chen & Meng, 2013). Furthermore, decent seed development and maturation are essential for seed vigor and longevity, despite the effects of environmental factors during seed development, such as temperature (Finch-Savage & Bassel, 2016; Sanhewe et al., 1996). Therefore, it is possible to identify the optimum harvest time through monitoring seed physiological indexes and changes in germination during different development stages. Lin et al. (2016a, 2016b) suggested that sheepgrass seed can be harvested at 39 days after peak anthesis and 10 days after the coloration of the lemmas. In our study, the results indicated that the seed water content continued to decrease during seed development and that the water content decreased significantly during 35–42 DAP. At 42 DAP, the seed water content decreased to approximately 15% (Fig. 2). In addition, our study found that 95% of the germplasm could germinate after 14 DAP and that the highest germination rate was observed from 35 DAP to 42 DAP in most of the germplasm (Fig. 3). Thus, our results suggested that the best harvest time for the different sheepgrass germplasm may be between 35 DAP and 42 DAP to ensure that the maximum germination ability of the seed has been acquired and to avoid seed shattering.

Primary dormancy is induced during the seed maturation phase and reaches a high level in freshly harvested seeds, and the level of dormancy slowly decreases during the subsequent dry storage of seeds (Graeber et al., 2012; Holdsworth, Bentsink & Soppe, 2008; Larson & Kellogg, 2009). Yi, Li & Tian (1997) suggested that sheepgrass undergo physiological dormancy and that the level of dormancy is the greatest in the physiological maturation period. Previous studies suggested that the ratio of ABA and GA plays important roles in the regulation of seed germination and dormancy (Holdsworth, Bentsink & Soppe, 2008). ABA induces seed maturation and promotes dormancy, and the seed ABA content of sheepgrass gradually increases from the milk to the ripening stage and then decreases from the ripening stage to the fully ripe period (Yi, Li & Tian, 1997). GA is required for seed germination, and it can act on reactive oxygen species to release dormancy and promote seed germination (Graeber et al., 2012); in sheepgrass seed, the content of GA continuously decreases after the filling stage, reaches a minimum before the ripening stage, and then begins to increase (Hu, Huang & Wang, 2012; Yi, Li & Tian, 1997). Previous studies showed that germination varied by germplasm in wheat (Mohsen, Mahdi & Abolfazl, 2011), rice (Bosetti et al., 2012; Gu et al., 2015), and spring barley (Briggs & Dunn, 2000), indicating the significant role of germplasm in determining seed vigor, germination and seedling growth. However, studies on the seed germination of sheepgrass were largely limited to only one or two germplasm, and the effect of germplasm on the seed germination of sheepgrass had not been identified. As multiple new sheepgrass cultivars were bred and conserved in recent decades, they have contributed to the investigations of the influence of germplasm on the variation in seed germination (Liu & Li, 2015). In this study, we used 20 germplasm of sheepgrass to investigate the germination characteristics during seed development, and the results showed that specific germplasm have representative germination characteristics that distinguish them from the other various sheepgrass germplasm. For example, S19 had a high germination rate, and S10 had a low germination rate (Fig. 3). These sheepgrass germplasm can be introduced as perfect candidates for further studies on seed dormancy and germination in sheepgrass. In addition, the seed germination patterns of eight germplasm of sheepgrass were highly similar between two consecutive years, suggesting that germplasm has a significant influence on the seed germination of sheepgrass (P < 0.01) (Fig. 4; Table 3).

Previous studies have suggested that the germination rate of sheepgrass seeds continuously increases during seed development and maturation (Lin et al., 2011) but was restrained by the germplasm of sheepgrass. Our results revealed a more comprehensive description of the changes in the germination rate of multiple germplasm during sheepgrass seed development, where six types of germination patterns during seed development were identified. For the different germplasm of sheepgrass, a decrease in the seed germination rate was observed during the seed development process, suggesting different variations in seed dormancy in different germplasm of sheepgrass (Fig. 4). The release of seed dormancy in germplasm such as S19 was persistent during seed development, resulting in a high germination rate for the mature seeds. In contrast, the germination rate of some germplasm, such as S10, remained relatively low throughout seed development, suggesting constant strong seed dormancy. In many other germplasms, the degree of seed dormancy varied among the different developmental stages, which requires further illumination. During late seed maturation and after-ripening, seed dormancy and germination characteristics may continue to change (Carrera et al., 2008; Haimovich et al., 2008). Veasey et al. (2004) observed a variation in the loss of dormancy of different rice populations and cultivars during after-ripening, resulting in different germination patterns. In Stipa bungeana, similar variations in seed germination among cultivars during after-ripening was identified (Zhang et al., 2017). In this study, we selected 10 germplasm of sheepgrass to reveal the after-ripening effect on seed germination, and four types of after-ripening effect patterns were observed (Fig. 5). Our results indicated that the after-ripening effect on seed germination of sheepgrass was significantly influenced by both germplasm and storage time. We observed the effect of after-ripening on sheepgrass within 2 months under the general storage conditions with the main purpose to verify the variation in different germplasm; however, the effects of storage time and multiple treatments of after-ripening on sheepgrass germination require further investigation.

Conclusion

This study shows that the best harvest time for different sheepgrass germplasm may be between 35 DAP and 42 DAP. Furthermore, the germination of sheepgrass seeds is significantly affected by the germplasm during both seed development and after-ripening. Hence, the germplasm of sheepgrass is the key factor determining seed germination and dormancy, and these results will lay a theoretical foundation for breeding new varieties of sheepgrass with high germination rates in the future.

Supplemental Information

Supplemental Information 1 Seed germination rate among different sheepgrass (Leymus chinensis) germplasms.

(A) Change in the weight and water content of the developing seeds of sheepgrass. (B) Six types of germination patterns during seed development. (C) The seed germination patterns of eight germplasms at different developmental stages during two consecutive years. (D) After-ripening effect on the seed germination rates of different sheepgrass germplasms after harvest.

Click here for additional data file.

Supplemental Information 2 Supplementary Fig. 1.

Population photo of sheepgrass at seed maturity period. Photograph by Gongshe Liu, Xiaoxia Li.

Click here for additional data file.

Additional Information and Declarations

Competing Interests

Author Contributions

Data Availability

The authors declare that they have no competing interests.

Weiguang Yang performed the experiments, analyzed the data, contributed reagents/materials/analysis tools, prepared figures and/or tables, authored or reviewed drafts of the paper.

Shu Liu performed the experiments, analyzed the data, contributed reagents/materials/analysis tools, prepared figures and/or tables, authored or reviewed drafts of the paper.

Guangxiao Yuan analyzed the data, contributed reagents/materials/analysis tools.

Panpan Liu analyzed the data, contributed reagents/materials/analysis tools.

Dongmei Qi contributed to the material planting, seed collection and preservation.

Xiaobing Dong contributed to the material planting, seed collection and preservation.

Hui Liu contributed to the material planting, seed collection and preservation.

Gongshe Liu conceived and designed the experiments, prepared figures and/or tables, authored or reviewed drafts of the paper, approved the final draft.

Xiaoxia Li conceived and designed the experiments, prepared figures and/or tables, authored or reviewed drafts of the paper, approved the final draft.

The following information was supplied regarding data availability:

The raw data are available in the Supplemental Files.

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
