# Peer review of "Germination characteristics among different sheepgrass (Leymus chinensis) germplasm during the seed development and after-ripening stages"

_PeerJ, doi:10.7717/peerj.6688_

## Round 0.1 · original submission · Major Revisions

This paper has now been seen by three reviewers. Although all reviewers agree that the topic could be of interest to the readership of PeerJ, all three reviewers have a number of issues that should be considered before the paper can be considered for publication in PeerJ. Therefore, I would suggest the authors to submit a revised version in which all issues brought up by the reviewers have been carefully addressed. Most likely, the revised version of the manuscript will have to be seen again by at least one of the reviewers.

·

Basic reporting

Basic reporting parameters are okay, but there are a few issues.

The authors need to define exactly what is meant by "genotypes", which is a term used in the title, abstract, and throughout the manuscript. Most or maybe all Leymus species are self-incompatible meaning that every plant and every seed is genetically unique. Thus, it is impossible to say that "All the genotypes of the seeds were obtained and harvested in both 2016 and 2017." (see lines 133-134). What exactly are the 20 genotypes mentioned on lines 131-132? Are these different collections from different locations or different plants from one population? Can the authors provide GPS coordinates for the origin of these population(s)?

I do not agree or understand the sentence on lines 38-29 "genotypes with typical germination patterns become ideal candidates for studying seed germination".

The word "great" was overused (three times) in the last two sentences of the first pargraph in Introduction (lines 51-54). The sentence on lines 50-53 has two redundant parts, it could be simplified "...grassland restoration."

Experimental design

The Materials and Methods for the after-ripening experiment was somewhat unclear. I understand that the authors measured germination of seeds from 20 genotypes harvested over six different stages of development, measured by days after pollination (DAP), and four different stages of after-ripening, with 3 replicates. Were there 6 DAP x 4 DAH treatments or 6 DAP + 4 DAH? I think it was 6 DAP + 4 DAH, where all DAH treatments were harvested at 42 DAP? If so, sentence 147-149 needs to be clarified and the authors should not say "each state of seed development as well as at during after-ripening process" (lines 51-152). Figure 5 shows four stages of after-ripening (0, 15, 30, and 60 DHA) but Table 4 only shows 2 df for after-ripening? I don't understand how the authors got 29 df for genotype in Table 4. It should be absolutely clear from the text in the Materials and Methods or Results how to explain df in all ANOVA tables.

Validity of the findings

The main conclusions of the study (lines 337-342) concern the importance of "genotype". However, the genotypes used in this study are not well-defined. In fact, the are not really defined at all. These genotypes could be different plants from the same population, different populations, or different plants from different populations? The readers really need more information about the "genotypes" used in this study.

Additional comments

Seed germination and harvest time are important topics for utilization of native perennial grasses, and I think that this Yang et al. PeerJ submission provides useful and practical information for Leymus chinensis.

Genotypic variation and QTLs controlling seed germination have also been documented in other Leymus species and we continue working on issues related to seed production and seed quality of native Leymus species of North America.

Larson, S.R., and E.A. Kellogg. Genetic dissection of seed production traits and identification of a major-effect seed retention QTL in hybrid Leymus (Triticeae) wildryes. Crop Sci. 49:29-40. 2009.

Best wishes.

Reviewer 2 ·

Basic reporting

.• The manuscript is well organized. Due to large number of datasets used, some places it became difficult to understand the correct explanation given by the authors.
• Besides some places; clear, unambiguous, professional English language used throughout.
• The introduction is reasonable and shows the context.
• Literature well referenced & relevant.
• Though the Figures are relevant, labelled & described but vertical axis mismatched in different sub-figures. The captions of the Figures need to be checked again.
• Data is robust, statistically sound, & controlled.
• Discussion and conclusion are linked to the original research question.

Experimental design

No comment

Validity of the findings

No comment

Additional comments

• Line number 16 and 44: Remove unnecessary brackets from (Leymus chinensis ((Trin.) Tzvel)) and change to (Leymus chinensis (Trin.) Tzvel).
• The following paragraph cited in the abstract is very long and creating confusion. This paragraph needs to be rephrased into two sentences for clear sound.
Line number 19-23. To clarify the appropriate seed harvest time and the effects of genotype, 20 seed development and after-ripening on seed germination, we selected 20 genotypes of 21 sheepgrass to analyze the seed morphological and physical changes as well as the seed 22 germination and dormancy characteristics of sheepgrass during seed development stages using a 23 7 - d gradient after pollination (DAP).
• Line number 23: Full form of DAP is not clear in the sentence.
• Line number 26-28: Furthermore, our results indicated that there were 6 types of germination patterns, including genotypes with increasing germination rates in the developing seed, such as S19 and Z37……………..in this sentence authors mentioned Z37 which is not correct as per results/Figure 3.
Explanation of Z37- the germination rate continued to increase at the early developmental stages until 35 DAP and then began to decrease.
• Line number 30: add the years before 2016 and 2017; result change to results.
• Line number 131: coded asK8; add space in between as and K8.
• Line number 134: add the years before 2016 and 2017.
• Line number 136: Why only 4 Genotypes i.e. K4, S19, S10, and N14 have been mentioned under heading Morphological and physical observations of the developing seed? Not clear from supplemental excel file too?
• Line number 161: Date analysis should be change as Data analysis.
• Line number 277: approximately15% add space in between.
• Figure 1: Authors are advised to provide a habit/population photo of Leymus chinensis which ensure a clear image of the plant to the international audience.
• Figure 3: Since the horizontal axis value are same for all the sub-figures, if possible try to keep the vertical axis value same for all sub-figures or in two groups such as A, C, E up to 100 and B, D, F up to 80.
• Figure 3: Authors missed caption (C) and in place of (C) authors have mentioned D. In same; caption D is not in the bracket. In the caption of Figure 3A, Genotype N14 is missing. In the caption of Figure 3B, Genotype C18 is missing.
• Figure 4A: Vertical axis value mismatched and giving wrong interpretation; keep the vertical axis value up to 100 as mentioned in 4B and 4C. In vertical axis add 14 DAP, 28 DAP and 42 DAP in sub-figures A, B, C respectively for the easy explanation as given in the caption.
• Figure 5: Not in a uniform way. Authors are advised to keep the vertical axis value same for all the figures such as 0 to 100. In Figure 5C the vertical axis value 100 is missing.
• Since the paper is characterizing seed germination, authors are advised to add types of germination (epigeal/hypogeal) in sheepgrass in the text.
• How the authors are confident that pollination has occurred in different Genotypes of sheepgrass? How the authors are confirmed about the exact date of pollination such as a first day or first few Days After Pollination?

·

Basic reporting

1. The paper is an outcome of a very simple experiment, the finding of which are worth a research note.The only information generated is that germination of seeds of different genotypes differ; the underlying reason for this behavior has not been investigated/explained in terms of, seed viability, type of seed dormancy etc.
This information will be of practical importance only if the results (days after pollination) are interpreted in terms of the physiological stages of the crop (leaves become straw colored, seeds turn cream to brown etc.) as pollination that spreads over a few days can not be made a criteria for deciding the harvesting time.

2. In the introduction, there is a mention of pre-sowing treatments that promote germination of seeds. But in the methodology there is no mention of the use of pre-sowing treatments.

4. References are absent in 'Materials and Methods'. For example ISTA manual is an important reference material in seed testing; this not mentioned.

5. The term '7-d gradient' in the Abstract, is ambiguous.

Experimental design

There is no mention of any design (Simple Random Design, Randomised Block Design etc)

Validity of the findings

No comments

Additional comments

No comments.

---

## Round 0.2 · accepted · Accept

The authors have done their utmost to address all issues raised by the reviewers and this paper can now be accepted for publication.

# As part of his evaluation, Dr Lazo noted the following edits which should be resolved while in production:

LINE 2: / germplasms / germplasm / [In MOST contexts the plural form of ‘germplasm’ IS ‘germplasm’. In this manuscript ‘germplasms’ is very much over-used. Perhaps change all ‘germplasms’ to ‘germplasm’ and use sparingly.]

Others similar to the issue on LINE2: LINES 2, 19, 20, 25, 27, 28, 30, 31, 34, 35, 77, 109, 118, 123, 128, 130, 147, 148, 150, 190, 191, 205, 206, 207, 209, 210, 212, 214, 230, 232, 234[x2], 245, 247, 248, 251, 267, 282, 283, 284, 303, 307, 308, 310, 311, 313, 319, 321, 322, 323, 325, 327, 334, 339, 343, Figure 3[x5], Figure 4, Figure 5[x4], Table 1, Table 2, Table 3, and Table 4. I think the word ‘germplasms’ is highly misused and should be put into the form ‘germplasm’. There may be a few instances where ‘germplasms’ might be used, but it would be better to have it in the ‘germplasm’ form. This may be a remnant from the earlier request to change the word from 'genotypes' and thus the word form was misrepresented.

LINE 25: / will be / suggested to be /
LINE 76/77: / aimed to / [Remove.]
LINE 152: / at stage / at the stage /
LINE 153: / during after-ripening / during the after-ripening /
LINE 158: / when they / when the /
LINE 164: / designs are / designs were of a /
LINE 188/189: / , and a population photo of sheepgrass at seed maturity period was in the supplementary Figure 1. / ; a population photo of sheepgrass at the seed maturity period is provided in supplementary Figure 1. /

·

Basic reporting

All basic reporting criteria are satisfactory.

Experimental design

Experimental design criteria are satisfactory.

Validity of the findings

good

Additional comments

good

·

Basic reporting

Acceptable

Experimental design

Sound

Validity of the findings

Valid

Additional comments

This revised draft is acceptable